# A Synergistic pH-Responsive Serum Albumin-Based Drug Delivery System Loaded with Doxorubicin and Pentacyclic Triterpene Betulinic Acid for Potential Treatment of NSCLC

**DOI:** 10.3390/biotech12010013

**Published:** 2023-01-26

**Authors:** Zally Torres-Martinez, Daraishka Pérez, Grace Torres, Sthephanie Estrada, Clarissa Correa, Natasha Mederos, Kimberly Velazquez, Betzaida Castillo, Kai Griebenow, Yamixa Delgado

**Affiliations:** 1Chemistry Department, University of Puerto Rico, Rio Piedras Campus, San Juan 00925, Puerto Rico; 2Neuroscience Department, Universidad Central del Caribe, Bayamon 00960, Puerto Rico; 3Biochemistry & Pharmacology Department, San Juan Bautista School of Medicine, Caguas 00727, Puerto Rico; 4Biology Department, University of Puerto Rico—Cayey, Cayey 00736, Puerto Rico; 5Chemistry Department, University of Puerto Rico—Humacao, Humacao 00727, Puerto Rico

**Keywords:** betulinic acid, bovine serum albumin, doxorubicin, drug delivery system, lung cancer, synergistic effect

## Abstract

Nanosized drug delivery systems (DDS) have been studied as a novel strategy against cancer due to their potential to simultaneously decrease drug inactivation and systemic toxicity and increase passive and/or active drug accumulation within the tumor(s). Triterpenes are plant-derived compounds with interesting therapeutic properties. Betulinic acid (BeA) is a pentacyclic triterpene that has great cytotoxic activity against different cancer types. Herein, we developed a nanosized protein-based DDS of bovine serum albumin (BSA) as the drug carrier combining two compounds, doxorubicin (Dox) and the triterpene BeA, using an oil-water-like micro-emulsion method. We used spectrophotometric assays to determine protein and drug concentrations in the DDS. The biophysical properties of these DDS were characterized using dynamic light scattering (DLS) and circular dichroism (CD) spectroscopy, confirming nanoparticle (NP) formation and drug loading into the protein structure, respectively. The encapsulation efficiency was 77% for Dox and 18% for BeA. More than 50% of both drugs were released within 24 h at pH 6.8, while less drug was released at pH 7.4 in this period. Co-incubation viability assays of Dox and BeA alone for 24 h demonstrated synergistic cytotoxic activity in the low μM range against non-small-cell lung carcinoma (NSCLC) A549 cells. Viability assays of the BSA-(Dox+BeA) DDS demonstrated a higher synergistic cytotoxic activity than the two drugs with no carrier. Moreover, confocal microscopy analysis confirmed the cellular internalization of the DDS and the accumulation of the Dox in the nucleus. We determined the mechanism of action of the BSA-(Dox+BeA) DDS, confirming S-phase cell cycle arrest, DNA damage, caspase cascade activation, and downregulation of epidermal growth factor receptor (EGFR) expression. This DDS has the potential to synergistically maximize the therapeutic effect of Dox and diminish chemoresistance induced by EGFR expression using a natural triterpene against NSCLC.

## 1. Introduction

For decades, chemotherapy has been the treatment of choice for many cancer patients. However, there is no gold-standard therapeutic approach to eradicate cancer. This prompts the development of new strategies for more specific treatment designs depending on the diagnosed cancer type [1]. Considering that chemotherapy affects all growing cells, lack of tumor specificity is one of the most significant drawbacks of cancer treatments due to its severe toxic side effects. Furthermore, acquired chemoresistance produces more aggressive cancer cells in patients undergoing treatment. The development of nanosized DDS has been an emerging strategy to treat cancer and other diseases that need more target-specific therapies [2,3,4].

Doxorubicin (Dox), a natural anthracycline, is a chemotherapeutic drug recommended for several types of cancer, e.g., small-cell lung carcinoma (SCLC) [2]. However, Dox is not curative to non-small-cell lung carcinoma (NSCLC) [3] and also induces resistance, diminishing its therapeutic effects [4]. The mechanism of action of Dox promotes the inhibition of the DNA topoisomerase in RNA/DNA synthesis and an increase in reactive oxygen species [5]. In addition, Dox also induces chemoresistance by activating the epidermal growth factor receptor (EGFR) [6]. Due to its lack of selectivity and chemoresistance, patients undergoing Dox treatment may experience cardiac toxicity and cancer recurrence [5,7]. 

Betulinic acid (BeA), the other compound investigated herein, is a lupine-type pentacyclic triterpene mainly isolated from birch trees. BeA is a phytochemical compound with great potential against several diseases. Some multifunctional aspects include its antimicrobial, antiviral, anti-inflammatory, and anticancer activities [8,9]. The anticancer properties of BeA have shown potent cytotoxic activity against various types of cancer in vitro and in vivo [10]. In addition, researchers have reported BeA’s antitumor mechanism to occur through mitochondrial oxidative stress induction, the regulation of SP transcription factor Sp1 (specificity protein 1, a protein encoded by SP1 gene), and the inhibition of proliferative factors mediating tumor cell death [10]. However, an essential disadvantage of BeA, which limits its biomedical use to a greater extent, is its poor water solubility of only 0.02 µg/mL at room temperature [8]. To improve this, researchers have studied different encapsulation methods, including self-assembling properties, to enhance BeA bioavailability under physiological conditions [11,12]. Thus, several benefits arise from drug combinations working in synergy, including decreased toxicity and fewer side effects compared with higher doses of single drugs [13,14]. Hence, combining BeA with other drugs could work as an excellent alternative anticancer treatment.

The most abundant plasma protein is serum albumin (SA), a highly water-soluble and stable protein over a wide pH range (from pH 4 to 9). SA possesses many interaction sites (e.g., pockets) and functional groups (e.g., amines and carboxyls) ideal for covalent modification and molecule- (polar and non-polar) and ion-loading [15]. Additionally, the flexibility of the SA structure facilitates the binding of different compounds [15]. Interestingly, SA has a primary and three-dimensional structure highly conserved in mammals, e.g., bovine SA has 76% amino acid similarity with human SA [16]. Human serum albumin (HSA) and bovine serum albumin (BSA)-based DDS have shown many advantages over the “naked” drugs, including a better NP size, homogeneity, storage, physiological stability, solubility and dispersion in aqueous solution, biocompatibility, biodegradability, and the possibility of surface modification leading to more specific therapy [17]. Moreover, cancer cells overexpress various albumin receptors (i.e., gp60 and SPARC), which can contribute to enhanced albumin-based nanoparticle uptake [18]. Specifically, Abraxane (albumin-based nanoparticles containing paclitaxel) shows improved drug efficiency over Taxol (paclitaxel) [19].

NSCLC, the predominant form of lung cancer, is a heterogeneous class of tumor, where the most common subtypes are adenocarcinoma, squamous cell carcinoma, and large-cell carcinoma, among other less frequent subtypes. This tumor is usually less sensitive to chemo- and radiotherapy and significantly affects human health [20]. Effective measures for cancer therapeutics are under evaluation, such as cyclodextrin [21], liposomal [22], carrier-free nanodrugs [23], and micelles [24]. However, a promising therapeutic approach still does not exist [25,26,27]. As an indispensable resource, an albumin-based drug delivery system has demonstrated potential value in countering NSCLC. In this work, we encapsulated a BeA and Dox drug combination in a BSA-based DDS using a cost-effective oil-in-water-like emulsion method. We characterized the resulting nanosized DDS and determined the drug release in tumor-like and normal physiological environments. The effect of the BEA, Dox-loaded DDS, and the isolated drugs on NSCLC A549 cell viability was measured and analyzed for synergistic interactions. Additionally, we investigated the drugs’ effect on the metabolic activity of the treated A549 cells.

## 2. Materials and Methods

### 2.1. Chemicals and Reagents

Fatty acid free-bovine serum albumin (BSA), betulinic acid (BeA, 90% purity), and doxorubicin (Dox) were purchased from Sigma-Aldrich (St. Louis, MO, USA). The cell lines A549 (human lung adenocarcinoma; ATCC CCL-185) and MRC5 (human fibroblast-like; ATCC CCL-171) were from the American Type Culture Collection (Manassas, VA, USA). CellTiter 96 aqueous non-radioactive cell proliferation assay (3-(4,5-dimethylthiazol-2-yl)-5-(3-carboxymethoxyphenyl)-2-(4-sulfophenyl)-2H-tetrazolium (MTS) reagent) was from Promega Corporation (Madison, WI, USA). NucBlue Fixed Cell Ready Probes Reagent (4′,6-diamidino-2-phenylindole dihydrochloride, DAPI), fluorescein (FITC), and Vybrant DiO Cell-Labeling Solution were purchased from Thermo-Fisher Scientific (Waltham, MA, USA). Cell cycle, EGFR expression, multi-caspase activation, DNA damage, and oxidative stress (ROS) assays were from Luminex Corporation (Austin, TX, USA). All other chemicals were purchased from various suppliers in analytical grade and used without further purification.

### 2.2. Preparation of the Nanosized DDS

Protein-based DDS were designed using an oil-in-water-like emulsion following a nanoprecipitation step. We obtained protein NPs following a procedure with several adaptations as described by Molina et al. 2016 and Delgado et al. 2015 [28,29]. DDS synthesis was optimized by adjusting several parameters. Briefly, BSA was dissolved in a 1× phosphate buffer solution, pH 7.4, to achieve a final concentration of 5 mg/mL at 40 °C. The two drugs, BeA and Dox, were dissolved in N, N-Dimethylformamide (DMF). Then, the solvent was added with a syringe needle at a constant flow of 120 mL/h to the BSA solution to achieve a final concentration ratio of 1:2 protein:drug (*w*:*w*). The dispersed emulsion was left stirring at a constant rate of 100 rpm for 4 h. Afterward, this preparation was centrifuged thrice with nanopure water using a 10 kDa filter unit at 5000 rpm for 10 min to remove the unattached free drug moieties. The final DDS solution was freeze-dried for 48 h and stored at −20 °C until further use.

### 2.3. Characterization of the DDS

#### 2.3.1. Quantification of the DDS Components

We used spectrophotometric measurements to determine each component’s concentration in the DDS. The standard protocol using the Bradford Coomassie reagent was utilized to determine the BSA protein concentration in the DDS from the absorbance at 570 nm [30]. A calibration curve was constructed with BSA standards. An aliquot of 5 µL of each sample and BSA standard was analyzed spectrophotometrically following the Bradford Coomassie protocol, which established that 150 µL of Bradford reagent must be added to each sample and standard. Then, the samples were incubated for 10 min at 25 °C in the dark and followed by absorbance measurement at 570 nm.

The Dox concentration was determined using its intrinsic absorbance at 485 nm [31]. For the determination of the BeA, we chose to use the vanillin-sulfuric acid assay [32]. In brief, we prepared several dilutions from a BeA stock solution in ethanol (5 mg/mL) for a calibration curve. The dilutions and the DDS samples were heated at 85 °C for solvent evaporation. Afterward, 250 µL of vanillin stock (50 mg/mL) was added, followed by the addition of 500 µL of sulfuric acid (95.0–98.0%) to each sample. Then, the solutions were heated at 60 °C for 30 min. Subsequently, the solutions were transferred into an ice bath. Then, 2 mL of acetic acid (99.7%) were added and incubated for 20 min. Once room temperature was reached, the DDS samples and triterpene dilutions were measured at 548 nm. For these colorimetric assays, we used a microplate reader spectrophotometer (Multiskan™ FC Microplate Photometer; Thermo Scientific™, Waltham, MA, USA).

#### 2.3.2. Encapsulation Efficiency (EE) and NP Yield 

An aliquot of 20 µL of the DDS was collected to determine the concentration of each drug. The final amount of each drug in the DDS was obtained as explained before for the quantification of Dox and BeA. The equation used to calculate this parameter is [1]:

% EE = ((final drug weight in DDS)/(Initial weight of drug for DDS preparation)) × 100


The nanoparticle yield was calculated by using the equation [18]:
NP yield: ((final amount (mg) of BSA in DDS)/ initial amount (mg) of BSA) × 100


#### 2.3.3. Particle Size, Polydispersity, and Zeta Potential

The DDS size, polydispersity index, and zeta potential were measured using a dynamic light scattering (DLS) instrument (Mobius model, Wyatt Technology, Santa Barbara, CA, USA). Samples were dissolved in nanopure water and placed in a quartz cuvette for the measurements. Each value is an average of three runs of a round comprised of 13 measurements each.

#### 2.3.4. In Vitro Drug Release Studies

Drug release experiments were conducted as described by Morales-Cruz et al. [33]. Briefly, one ml of a 5 mg/mL solution of the DDS in 25 mM sodium bicarbonate at pH 6.8 and 7.4 (to mimic tumor microenvironment and physiological conditions, respectively) was incubated at 37 °C under stirring. At pre-determined time intervals (typically 24 h), the DDS sample was filtered through a centrifugal filter device at 12,500 rpm for 15 min. The filtrated solution (with the released drug) was measured at 470 nm (Dox intrinsic absorbance) after the filtration through the centrifugal filter device. Additionally, the BeA component was quantified from the filtered solution using the vanillin-sulfuric acid assay as described in Section 2.3.1. The DDS was re-suspended in 1 mL of bicarbonate buffer to maintain sink release conditions. The data are presented as accumulative release profiles of an average triplicate, with standard deviations.

### 2.4. Cell Culture Experiments

#### 2.4.1. Cell Culture Conditions

NSCLC A549 and human normal lung MRC5 cell lines were maintained following ATCC protocols. These cells were cultured in supplemented Dulbecco’s Modified Eagle’s Medium (DMEM, containing 1% L-glutamine, 10% fetal bovine serum (FBS), and 1% penicillin/streptomycin). Cells were kept in a humidified incubator under 5% CO_2_ and 95% air at 37 °C. We conducted all the experiments before the cells reached 25 passages.

#### 2.4.2. Cell Viability

A549 and MRC5 cells were seeded into 96-well plates at a density of 1 × 10^5^ cells/well in 0.1 mL supplemented DMEM medium. After 24 h, cells were treated with various concentrations (10, 25, 50, 75, 100 mg/mL) of BSA-(Dox+BeA) DDS and controls (BSA DDS, BSA-BeA, BSA-Dox) and incubated for 24 h. Then, 10 uL of the MTS reagent from CellTiter 96^®^ Aqueous Non-Radioactive Cell Proliferation Assay (Promega G5421) was added to each well. Then, we incubated the plate at 37 °C and 5% CO_2_ atmosphere for 1 h. After incubation, absorbance was measured at 492 nm in a microplate reader (mean ± SD, n = 8). The cell viability was calculated as follows:

% Viability = (sample data − incubation media data)/(untreated cells data − incubation media data) × 100


The 50% inhibitory concentration (IC_50_) was calculated by Graphpad Prism 9 software (Prism 9; GraphPad by Dotmatics, San Diego, CA, USA).

### 2.5. Synergism, Additive or Antagonistic Drug Interactions

We measured A549 cell viability with different concentrations of Dox and BeA by the MTS reagent to evaluate the cytotoxic interactions of the BeA and Dox combination. The absorbance was measured at 492 nm after 1 h of incubation. The cell viability was calculated from a quadruplicate sample group. The concentration index (CI) was calculated by CompuSyn software (www.combosyn.com, accessed on 7 October 2022) based on the Chou and Talalay [34] equation.
CI = D1/Dx1 + D2/Dx2

In the formula, Dx1 or Dx2 is the drug dose alone with the inhibition on x%, and D1 or D2 is the portion of the drug in combination with the same inhibition on x%. After the interaction of different drugs is determined, the CI value can be obtained. Depending on the CI, the drug interactions can be described as synergism, antagonism, or an additive effect. A CI of 1 indicates additive effects in the drug combination studied; a CI of <1 indicates synergism, while a CI of >1 indicates antagonism.

In addition, we used Synergy Finder 2.0 software (https://synergyfinder.fimm.fi/, accessed on 7 October 2022) to determine a synergy score for combination therapy [35]. Synergy Finder 2.0 is an open and free software that implements an efficient synergy estimation for multidrug combinations, novel visualization, and statistical treatment of replicate measurements, among other features, to determine synergism in drug combinations. When the synergy score is larger than 10, it indicates a synergistic effect; if the score is less than −10, it indicates an antagonistic effect; if the score is between −10 and 10, the drug interaction is additive.

### 2.6. Cellular Internalization of DDS

The DDS were labeled with fluorescein isothiocyanate (FITC) to visualize cellular internalization using confocal microscopy (Eclipse Ti-E Inverted Fluorescence Microscope; Nikon Healthcare Business Division, Tokyo, Japan).

#### 2.6.1. DDS-FITC Labeling

Controls (BSA) and samples (BSA-(Dox+BeA) DDS) were labeled with FITC by 24 h of incubation. In brief, the DDS was dispersed at 2 mg/mL BSA concentration in 0.1 mM sodium bicarbonate (pH 7.4) buffer, and then 50 µL of FITC (1 mg/mL FITC stock in DMSO) was gently added per 1 mL of protein solution. The reaction was performed under darkness with continuous stirring for 8 h at 4 °C. We centrifuged the reaction mixture using a filter device (10 kDa cut off) at 5000 rpm for 10 min to remove the excess FITC. Then, we repeated the centrifugation with nanopure water for up to five washes. The labeled DDS were freeze-dried for further use.

#### 2.6.2. DDS Uptake Visualization

A549 cells were seeded in 8-well cover slip plates with a density of 10,000 cells/well and incubated for 24 h in supplemented culture media (DMEM). Then, A549 cells were incubated with DDS and controls for 24 h. Afterward, the A549 cells were washed with PBS 1X and fixed for 15 min with a 3.7% formaldehyde solution at 37 °C. After removing the fixing solution, wells were washed twice with PBS 1×, followed by incubation at room temperature for 15 min of DAPI nuclear stain and Vybrant DiO dye solutions. Plates were stored at 4 °C until analysis. The plates were observed by confocal microscopy using a Nikon Eclipse Ti microscope and 60× objective. FITC was excited at 487 nm and monitored at 525 nm; DAPI was excited at 402 nm, and the emission was observed at 420–480 nm; Vybrant DiO was excited at 484 nm, and the emission was observed at 501 nm. Fluorescence intensity was analyzed using the NIS-Elements Viewer program (version 5.21 64-bit; Nikon Healthcare Business Division, Tokyo, Japan).

### 2.7. Flow Cytometry Analysis

We used flow cytometry to analyze several cellular events involving metabolic pathways. In general, cells were seeded in a 6-well plate at a density of 10,000 cells/well and incubated for 24 h in DMEM medium containing L-glutamine, 10% FBS, and 1% penicillin/streptomycin at 37 °C in 5% CO_2_. After 24 h, A549 cells were treated with DDS samples and controls, followed by each manufacturer protocol. The flow cytometer instrument used was a Muse Cell Analyzer (Muse® Cell Analyzer; EMD Millipore Corporation, Temecula, CA, USA). 

#### 2.7.1. Cell Cycle Arrest

For the cell cycle assay (Luminex MCH100106), we followed the manufacturer’s protocol. Briefly, after A549 cells were incubated with treatments for 24 h, the medium was discarded, and cells were scraped and centrifuged. The pellet was mixed with cold 70% ethanol and centrifuged for fixing. Afterward, the pellet was washed, suspended in the cell cycle reagent, incubated for 30 min, and measured after mixing.

#### 2.7.2. Multi-Caspase Activation

The multi-caspase assay (Luminex MCH100109) was performed following the manufacturer’s protocol. After the treatment incubation, A549 cells were scraped and centrifuged. The pellet was dissolved in the assay buffer provided by the kit, and the working solution was incubated for 30 min at 37 °C in a 5% CO_2_ atmosphere. Afterward, a caspase working solution was added and incubated for 5 min at room temperature for measurement.

#### 2.7.3. DNA Damage Induction

The DNA damage assay (Luminex MCH200107) was performed following the manufacturer’s protocol. After the treatment incubation, A549 cells were scraped and centrifuged. The pellet was dissolved in the assay buffer provided by the kit. Later, fixation buffer was added and incubated for 10 min. Afterward, the centrifuged pellet was dissolved in permeabilization buffer (also provided with the kit) and incubated for another 10 min. Then, the permeabilization buffer was discarded after centrifugation, and the antibody cocktail was added and incubated for 30 min. Next, the solution was washed with assay buffer and suspended in fresh assay buffer for measuring.

#### 2.7.4. Oxidative Stress Production

The oxidative stress assay (Luminex MCH100111) was performed following the manufacturer’s protocol. After A549 cells were treated for 24 h, the cells were scraped and centrifuged to add the reagents from the kit, incubated for 30 min, and washed to be measured in the cell analyzer.

#### 2.7.5. EGFR Expression

The EGFR expression assay (Luminex MCH200102) was performed following the manufacturer’s protocol. After the treatment incubation, A549 cells were scraped and centrifuged. The pellet was dissolved in the assay buffer provided in the kit. Later, the fixation buffer was added and incubated for 5 min. Afterward, the centrifuged pellet was dissolved in permeabilization buffer (also provided by the kit) and incubated for another 5 min. The permeabilization buffer was discarded after centrifugation, and the antibody cocktail was added and incubated for 30 min. Next, the solution was washed with assay buffer and suspended in fresh assay buffer to measure the EGFR expression.

### 2.8. Statistical Analysis

All experiments were performed at least in triplicate. All data were expressed by plotting values with an average of four to eight measurements for each treatment condition as mean ± standard deviation (SD). Quantitative data were analyzed with the statistical software GraphPad Prism 9 ((Prism 9; GraphPad by Dotmatics, San Diego, CA, USA). Statistical analysis was performed using one-way analysis of variance (ANOVA) to compare the mean value of each condition versus the control. The P values levels of statistical significance defined by the statistical software Prism are; exorbitantly significant (****) when value is lower than 0.0001, extremely significant (***) when values are from 0.001 to 0.0001, very significant (**) when values are from 0.001 to 0.01, significant (*) when values are from 0.01 to 0.05, and non-significant (n.s.) when values are equal or greater than 0.05.

## 3. Results

### 3.1. Preparation and Characterization of BSA-(BeA+Dox) DDS

The protein-based DDS NPs were prepared using an oil-in-water-like emulsion system where the drugs were dissolved in the organic solvent DMF and added to an aqueous BSA solution. The hydrophobicity of BeA contributed to the self-assembly formation of the DDS. Many conditions were varied, such as protein-drug ratios, incubation times, and whether to incorporate a lipid coating to improve nanoparticle stability (Figure 1). Also our DDS was prepared and characterized with just one drug (Dox or BeA) to compare with the DDS loaded with two drugs.

#### 3.1.1. Size and Charge of the DDS

The properties of the DDS were characterized to identify the best conditions in terms of size and polydispersity. These properties are important for the therapeutic success and tumor accumulation of any DDS directed toward solid tumors [36]. The diameter size, surface charge (z-potential), and polydispersity index of the DDS particle were determined by DLS (Table 1). All developed DDS showed particle sizes in the nanometer range. The BSA-BeA DDS showed the smallest diameter (size: 97 ± 1 nm, polydispersity: 27 ± 2%), followed by BSA-Dox (size: 138 ± 13 nm, polydispersity: 51 ± 14%) and BSA-(Dox+BeA) (size: 181 ± 2 nm, polydispersity: 23.2 ± 0.4%).

The surface charge is measured using the Zeta potential of the NP and is relevant to the dispersion stability and cellular internalization of the DDS [37]. The Zeta potential value can be related to particle stability due to the representation of the electrostatic force around the particle and the tendency to aggregation [38,39]. Even when positive charged NPs are favored for higher internalization in the literature, there are many studies using neutral and negative NPs that have shown high internalization [40,41]. The Zeta potential for all the BSA DDS with one drug (BSA-BeA: −4.6 ± 0.4, BSA-Dox: −3 ± 1) and with two drugs (BSA-(BeA+Dox): −2.7 ± 0.7) were quite similar.

#### 3.1.2. Encapsulation Efficiency (EE) of BeA and Dox

We implemented a vanillin-sulfuric acid method to quantify the total amount of the triterpene BeA in the DDS. The concentration of BSA and Dox in the DDS was determined using a Bradford assay and Dox intrinsic absorbance at 485 nm, respectively (Table 2). Those values were used to calculate the amount of each component in the DDS and, subsequently, the % EE for each drug and carrier yield (Table 3). The BSA-BeA DDS contained 18 ± 6 µM of BeA and 77 ± 11 µM of BSA. BSA-Dox DDS had 43 ± 6 µM of Dox and 131 ± 24 µM of BSA. Remarkably, BSA-(BeA+Dox) showed the highest concentration for both drugs (61 ± 6 µM Dox and 27 ± 14 µM BeA) in 110 ± 3 µM of BSA in comparison to one-drug component DDS. Furthermore, these quantifications were used to obtain the drug EE in the NPs. The tendency observed in the drug concentrations in each DDS was very similar for the EE and the carrier yield, where BSA-(BeA+Dox) showed the highest %EE (18 ± 4% for BeA and 77 ± 15% for Dox) and carrier yield (80 ± 12%).

#### 3.1.3. Circular Dichroism (CD) Analysis

To determine the perturbation on the structure of BSA in the formation of the DDS NPs, its secondary structure was observed through circular dichroism (CD). Additionally, this analysis can be used to confirm the loading of the drugs (BeA and Dox) into the BSA cavities throughout secondary structure changes. The structural patterns of native BSA and after drug (BeA and Dox) loading were determined and shown in Figure 1. All the samples were analyzed using the same concentration (0.5 mg/mL). For the secondary structure, we can appreciate a similar tendency pattern but with less intensity. This decrease is more notorious for the DDS loading two drugs. The change in secondary structure spectra (190–250 nm) increases when BeA and Dox are bound to BSA. At 200 nm, the absorbance decreases, corresponding to the protein backbone at the absorption of a cyclic ring. This decrease occurs because of configurational changes due to the increase in α-helix content [42].

#### 3.1.4. Cumulative Drug Release Profile

In a DDS, the drug release responding to an external stimulus is a key property to increase the drug specificity and decrease toxic side effects [43]. In this way, the cumulative Dox and BeA release profile was acquired by incubating our DDS and loading the two drugs at 37 °C in the most abundant physiological buffer, sodium bicarbonate (25 mM) at normal plasma pH 7.4 and at a more acidic pH 6.8 to mimic the tumor microenvironment. Figure 2 shows the results of the cumulative release. Our system (BSA-(BeA+Dox)) at pH 7.4 showed a low initial burst (~16% for both drugs) in the first hour (Figure 2A,B zoom inset pH 7.4). The initial burst at pH 6.8 was drastically increased for BeA, while it remained quite similar for Dox (Figure 2A,B zoom inset pH 6.8). At pH 6.8, 52.4 ± 0.7% of Dox and 62 ± 1% of BeA were released within 24 h. A release in which the DDS can accumulate in the tumor, followed by fast drug release, is desired. At pH 7.4, the release was significantly slower, and within 24 h, 28.9 ± 0.7% of Dox was released, as well as 38.8 ± 0.4% of BeA. At pH 6.8, both agents were fully released from the DDS after 72 h. In contrast, at pH 7.4, BeA and Dox were fully released at 96 h and 144 h, respectively. A faster release in the tumor microenvironment is a desired property.

### 3.2. In Vitro Assays

The in vitro assays were performed with A549 cells. We used MRC5 fibroblast cells to allow for comparison with normal non-cancerous cells and only used A549 in the mechanistic assays. The incubation time was 24 h, and the drug concentrations employed were the initially determined IC_50_ values. Native fatty acid-free BSA was used as a negative control for all the in vitro assays.

#### 3.2.1. Cytotoxicity of BeA and Dox in A549 Cells

A dose-response curve was created for A549 cells to confirm each drug’s ability to induce cell death. Each drug was incubated at several μM concentrations for 24 h in a confluency of 1 × 10^4^ cells/well to obtain the IC_50_ using an MTS viability assay. The IC_50_ values obtained were 98 ± 18 µM for Dox and 42 ± 2 µM for BeA (Figure 3). As expected, cell viability decreases at increasing drug concentration. Then, A549 cells were co-incubated with both drugs, and we observed a slight decrease (25 ± 2 µM BeA and 86 ± 1 µM Dox) in the IC_50_ concentration of each drug from this Dox + BeA combination (Figure 4).

#### 3.2.2. Molecular Interaction Effect of the BeA and Dox Combination

Because we determined a decrease in the IC_50_ of the drugs BeA and Dox combined, we decided to use the CompuSyn software to quantitatively analyze the molecular interaction effect of BeA and Dox and calculate the combination index (CI) values using the Chou and Talalay equation [34]. Figure 5A presents the CI results from CompuSyn. The mean CI value was 0.73 ± 0.2, indicating a synergistic effect (<1) of the drug combination against A549 cells. We also used Synergy Finder 2.0, a web application for the analysis of drug combination screening [35], to calculate the synergy score of the drug combination to confirm the results from CompuSyn. The synergy score was 19.06 (where >10 indicates synergistic effect) (Figure 5). This score can be interpreted as the average excess response due to drug interactions (i.e., a synergy score of 19.06 corresponds to 19.05% of response beyond expectation), revealing that the combination of BeA and Dox exhibits a synergistic effect on the viability of A549 cells. The more synergistic area score is 34.46, shown in the gray highlighted square of Figure 5B, when the concentrations are 40 µM and 100 µM for BeA and Dox, respectively.

#### 3.2.3. Cytotoxicity of the BSA-(BeA+Dox) DDS

After confirming that the combination of BeA and Dox worked synergistically together, we tested our BSA-(BeA+Dox) DDS to determine its effect against NSCLC A549 cells. Once the DDS was prepared and characterized, A549 cells were treated with a range of concentrations for 24 h and 48 h (Figure 6). The highest concentration of drugs loaded in the DDS (25.5 µM Dox and 13.0 µM BeA) reduced the viability to 42 ± 2% after 24 h and to 5.9 ± 0.8% after 48 h of treatment. Normal lung fibroblast MRC5 cells were also treated with the DDS. Cell viability was 58.0 ± 0.4% after 24 h and 0.5 ± 0.4% after 48 h. This means that, when employing isolated cells, the drugs impact any growing cell, as expected. However, one has to keep in mind that the passive accumulation of the nanosized DDS under in vivo conditions should reduce the impact on healthy cells in real treatment. This concept, in general, is the idea of nanosized systems loaded with cytotoxic drugs.

Then, the IC_50_ values of each drug in the DDS were calculated. When the DDS contains 13 ± 3 µM of BeA and 27 ± 6 µM of Dox, the cell viability of A549 cells reached 50%. This implies that the IC_50_ of BeA and Dox are significantly lower in the BSA-(BeA+Dox) DDS than the naked drugs individually and in combination. In addition, when MRC5 cells were treated with BSA-(BeA+Dox) DDS, we observed a slightly less cytotoxic effect compared to A549 cancer cells after 24 h incubation but an even larger effect after 48 h (Figure 6).

#### 3.2.4. Cellular Internalization of the BSA(BeA+Dox) DDS

A549 cells were grown on chambered slides and incubated with the FITC-labeled DDS for 24 h to trace the specific cellular internalization of the system. DAPI dye (blue fluorescence) was used to stain cell nuclei. Dox intrinsic fluorescence was observed as red fluorescence, and Vybrant Dio dye (green fluorescence) was used to stain the cellular membrane. After the A549 cells were treated with the samples, considerable amounts of BSA-(BeA+Dox) were internalized and observed in the membrane areas and the cell nucleus, respectively (Figure 7B). In addition, as a DNA intercalator, Dox was colocalized with cell nuclei [5] (Figure 7B,D).

In Figure 7B,D, where the cells were treated with BSA-(BeA+Dox), quenching of the blue and red dyes is expected due to Dox and DAPI binding to similar areas of the DNA in the nucleus [44].

### 3.3. Flow Cytometry Analysis

Using the IC_50_ of the drugs delivered via DDS as a reference, we chose to use the preparation that contains 5.2 μM of BeA and 10.2 μM of Dox (~IC_50_/2) at 24 h of incubation in A549 cells for the remainder of the mechanistic experiments. As a negative control, we used 50 μM fatty acid-depleted-BSA. As positive controls, we used the free drugs at 20 μM of BeA and 20 μM of Dox.

#### 3.3.1. Effect of BSA-(BeA+Dox) DDS on the Cell Cycle

The cell cycle is an ordered sequence of events to prepare for cell growth and division. In G0, a cell is in a resting or quiescent stage, and then, it can enter the cell cycle phases, where the cell has a size increase (G1 phase), synthesizes DNA (S-phase), synthesize proteins for cell division (G2-phase), and finally, divides by mitosis (M-phase) [45]. Figure 8 summarizes the results of the cell cycle analysis by flow cytometry. Treatment with 20 μM of free BeA did not cause significant changes in the cell cycle phases compared to untreated cells, with the highest population in the G0/G1 phase. This could mean that BeA induced cell cycle arrest at the G0/G1 phase, as demonstrated in the literature [46]. Meanwhile, Dox showed 51.4% G0/G1, 44.0% S, and 4.3% G2/M, showing a marked increase in cells in the S-phase. Exposure of the cells to the BSA-(BeA+Dox) DDS caused an even more remarkable increase in cells in the S-phase (50.9%). From the literature, Dox revealed cell proliferation inhibition through cell cycle arrest at the S- and G2/M- phases in a kidney cell line [47,48].

#### 3.3.2. Effect of BSA-(BeA+Dox) on Caspase Activity

Caspases are cysteine proteases that execute apoptosis (intrinsic and extrinsic programmed cell death). Dox can induce apoptosis through caspase activation [49,50]. Untreated A549 cells registered a total caspase activation of 57.2%. Treating the cells with the free drugs caused 68.3% activation in the case of Bea and 97.4% activation in the case of Dox 97.4%, respectively. An even higher level of caspase activation (99.3%) was caused by exposure to the BSA-(BeA+Dox) DDS. Comparison of the DDS with the free anticancer drugs (BeA and Dox) revealed a slight advantage of the DDS in caspase activation and cell death (Figure 9).

#### 3.3.3. Effect of BSA-(BeA+Dox) on the DNA Processing Machinery

DNA damage frequently affects the function of genes encoded [51]. DNA repair mechanisms are activated in the affected cell to correct the problem. Nevertheless, after a double-strand break, when damage might be too severe for repair, phosphorylation of the histone variant X (H2A.X) and ataxia telangiectasia mutated (ATM) kinase is promoted, leading to cell death [52]. The assay to assess the magnitude of this pathway indirectly measures double-strand breaks by measuring the amount of phosphorylated H2A.X and ATM. We found that Dox induced an increase in total DNA damage (97.04% vs. 20.85% in untreated cells), while the BSA-(BeA+Dox) DDS induced 92.48% (Figure 10). This result confirms the presence and functionality of Dox in the DDS. BeA did not show any increase in DNA damage.

#### 3.3.4. Effect of BSA-(BeA+Dox) on Oxidative Stress Production

Perturbation in redox homeostasis can increase the concentration of reactive oxygen species (ROS), which subsequently can cause cancer development. However, drastic increments in ROS could also promote cell death [53]. Cellular ROS production was measured by intracellular detection of hydroxyl (HO*) and superoxide (O_2_*) free radicals in A549 cells (Figure 11). BeA- and Dox-treated cells exhibited ROS production of 27.4% and 49.20%, respectively, compared with untreated cells. In addition, BSA-(BeA+Dox) promoted a high ROS production of 49.60%.

#### 3.3.5. Effect of BSA-(BeA+Dox) on Epidermal Growth Factor Receptor (EGFR) Expression

EGFR is a transmembrane glycoprotein that is overexpressed in several cancers, especially NSCLC, and it promotes cell proliferation, invasion, and chemoresistance [54]. In this study, we identified EGFR expression by flow cytometry and showed the non-expressing and expressing cells (inactivated and activated via phosphorylation). The untreated cells produced 24.0% of EGFR, whereas BeA- and Dox-treated cells produced 22.2% and 80.8%, respectively. Treatment of the A459 cells with the BSA-(BeA+Dox) DDS produced 41.7% of EGFR. We conclude that EGFR expression was significantly decreased when BeA was combined with Dox in the BSA-(BeA+Dox) DDS (Figure 12).

## 4. Discussion

Our system, BSA-(BeA+Dox), possesses properties (181 ± 2 nm of diameter size and −2.1 ± 0.7 mV of surface charge) that would allow for passive delivery and accumulation in tumors. We believe a possible mechanism is through the enhanced permeation and retention (EPR) effect caused by irregular tumor vasculature [20]. Furthermore, our DDS is a self-assembling complex between BSA and two molecularly different drugs (BeA and Dox). The triterpene BeA and the anthracycline Dox exposed strong interactions with the BSA structure pockets. Furthermore, the encapsulation efficiency (EE) was better when both drugs were loaded into the protein than just one drug alone.

Previous studies have developed albumin-based NPs into DDS [55,56]. In addition, structural studies of BSA using molecular dynamic simulation methods revealed how BeA [57] and Dox [58] are bound to BSA. BSA possesses large hydrophobic binding pockets on its surface. BeA binds to these large hydrophobic cavities of drug binding site I of subdomain IIA and IIB through hydrophobic and hydrogen bonding interactions [57]. The other drug, Dox, strongly binds to BSA by hydrophobic, hydrophilic, and hydrogen bonding interactions for stabilization. Those binding interactions alter the protein’s secondary structure, causing partial protein destabilization [57,58], which was registered in the CD analysis.

The term combination therapy is used when two or more drugs positively affect the drugs’ pharmacodynamics interaction (i.e., additive and synergistic). The primary purpose of combination therapy is to lower drug doses so that the patients experience fewer side effects while the efficiency of the treatment is enhanced [59]. Furthermore, repurposing old drugs with new applications as a potential treatment for other diseases, such as lung cancer, is advantageous to the drug combination [60]. In recent studies, drug combination therapies incorporating a natural-derived product have shown beneficial effects against NSCLC [61]. The results included cell viability reduction, apoptosis promotion, and protein expression downregulation to delay chemoresistance occurrence in lung cancer [62] and inhibition of cancer cell proliferation [63]. In our experiments, a synergistic effect is revealed in the drug combination of Dox and BeA, increasing the cytotoxic effect. This synergy was demonstrated by co-incubating A459 cells with free Dox and BeA, and the assistance of the nanosized BSA carrier increased this effect.

Considering that the extracellular pH in healthy tissue is 7.4, the pH in cancer tissues is more acidic (between 6.3 and 7.0) due to the dysregulation of acid-base homeostasis in tumors [64]. BSA-(BeA+Dox) DDS released the drugs faster in an acidic environment (pH 6.8) than in physiological conditions environments (pH 7.4), revealed by in vitro cumulative drug release studies adding some selectivity by this stimulus to the DDS. Under acidic conditions, the full release took 72 h, and after 24 h, the drug release was slightly higher than 50%. This tendency was also observed in cell viability studies on A549 cells at 24 h and 48 h. Confocal microscopy validated that the BSA-(BeA+Dox) DDS was internalized, and Dox was localized in the nucleus.

Normal cells, MRC5, were also treated with BSA-(BeA+Dox) DDS. The BSA-(BeA+Dox) DDS was more aggressive on cancer cells (A549) than normal cells (MRC5) after 24 h of treatment. However, when the treatment was extended for up to 48 h, the normal lung MRC5 cells also exhibited high mortality in the highest drug concentration. An important consideration about MRC5 fibroblast cells is that MRC5 are cells that enhance the invasive migration of cancer cells, promoting the Warburg effect and contributing to chemoresistance. The Warburg effect is characterized by the intensification of aerobic usage of glucose and/or lactate, promoting cancer reinforcement [65,66]. Thus, cytotoxicity to MRC5 after 48 h of treatment might provide an additional support to reduce chemoresistance.

The results from the cellular metabolic studies revealed that BSA-(BeA+Dox) produces S-phase cell cycle arrest, which was confirmed by the increase in DNA damage. The increase in oxidative stress through ROS could induce DNA disruption and/or mitochondrial membrane permeabilization. As a result, we found an increase in caspase activation, confirming the induction of apoptotic pathways. Interestingly, EGFR expression significantly decreased after DDS treatment, contrary to Dox alone. The BSA-(BeA+Dox) DDS might provide an alternative to diminish and evade multidrug chemoresistance in lung cancer patients.

## 5. Conclusions

In this study, we developed novel BSA-based NPs with the capacity to incorporate lipophilic (i.e., BeA) and hydrophilic moieties (i.e., Dox). The BSA-(BeA+Dox) DDS has the characteristics of size and charge (181 ± 2 nm and −2.1 ± 0.7 mV, respectively) suitable for passive delivery into lung tumors and to be internalized by NSCLC cells. The surface charge of the DDS NP was slightly negative, and this property has been shown to prolong circulating half-life and accumulation in tumors. Besides the synergistic effect of the BeA and Dox combination, the BSA-(BeA+Dox) DDS diminished to half the expression of EGFR commonly produced in NSCLC by chemotherapeutic agents such as Dox. BSA-(BeA+Dox) has the potential to increase the bioavailability of Dox and BeA and decrease chemoresistance during intravenous administration. Further studies of drug combination DDS must be continued to explore its anticancer efficacy for in vivo studies.

## Data Availability

Data is available upon reasonable request.

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
