# Peer review of "A Synergistic pH-Responsive Serum Albumin-Based Drug Delivery System Loaded with Doxorubicin and Pentacyclic Triterpene Betulinic Acid for Potential Treatment of NSCLC"

_biotech, 2023, doi:10.3390/biotech12010013_

Round 1

Reviewer 1 Report

pH-responsive serum albumin-based drug delivery system for treating NSCLC is particularly advantageou. The article by Torres-Martinez at al. is devoted to an important topic. I recommend this paper to be published in the journal. Here are some minor suggestions:

1: To be complete, the authors should enrich the related illustration about the article on NSCLC in “Introduction”. I also suggest that the authors to quote some latest results.

2: What motivated the authors to prepare the article. It is suggested to add some background and highlight the novelty of this work clearly. For example, “NSCLC, the predominant form of lung cancer, significantly impacts human health. Effective measures, such as cyclodextrin (Int. J. Mol. Sci. 2021, 22, 4783), liposomal (Pharmaceutics 2020, 12, 939), carrier-free nanodrugs (New J. Chem. 2022, 46, 17673-17677), and micelles (Nanoscale Res. Lett. 2021, 16, 124) are greatly needed in cancer therapeutics. However, promising therapeutic approach still do not exist (ChemPlusChem 2020, 85, 2143-2157; Biomedicines 2021, 9, 689; Cancers 2021, 13, 2139). As an indispensable resource, albumin-based drug delivery system has demonstrated potential value in countering NSCLC.” This is critical to address in this manuscript, the authors should enrich this part in the revised version.

3: An additional check of the language would be beneficial.

4: More recent research information and references on “combination drug therapies” should be added.

Author Response

The authors acknowledge and thank for all the comments and suggestions made by both reviewers. The incorporation of these changes helped us strengthen and clarify the article. All the changes made by the authors were highlighted in yellow for the convenience of the reviewers. We answered what we have done in each reviewer’s comment, referring to the sections and parts previously commented on by reviewers.

Reviewer #1

(Reviewer’s comments in italics for better understanding)

pH-responsive serum albumin-based drug delivery system for treating NSCLC is particularly advantageous. The article by Torres-Martinez et al. is devoted to an important topic. I recommend that this paper be published in the journal. Here are some minor suggestions:

1: To be complete, the authors should enrich the related illustration about the article on NSCLC in the “Introduction”. I also suggest that the authors quote some latest results.

Authors’ answer: We thank the reviewer for suggesting these changes. The paragraph now reads as follows: NSCLC, the predominant form of lung cancer, is a heterogeneous class of tumor, where the most common subtypes are adenocarcinoma, squamous cell carcinoma, and large cell carcinoma, among other less frequent subtypes. This tumor is usually less sensitive to chemo- and radiotherapy and significantly affects human health [20]. Effective measures for cancer therapeutics are under evaluation, such as cyclodextrin [21], liposomal [22], carrier-free nanodrugs [23], and micelles [24]. However, a promising therapeutic approach still does not exist [25-27]. As an indispensable resource, an albumin-based drug delivery system has demonstrated potential value in countering NSCLC.

Refer to lines 89-98.

2: What motivated the authors to prepare the article. It is suggested to add some background and highlight the novelty of this work clearly. For example, “NSCLC, the predominant form of lung cancer, significantly impacts human health. Effective measures, such as cyclodextrin (Int. J. Mol. Sci. 2021, 22, 4783), liposomal (Pharmaceutics 2020, 12, 939), carrier-free nanodrugs (New J. Chem. 2022, 46, 17673-17677), and micelles (Nanoscale Res. Lett. 2021, 16, 124) are greatly needed in cancer therapeutics. However, promising therapeutic approach still do not exist (ChemPlusChem 2020, 85, 2143-2157; Biomedicines 2021, 9, 689; Cancers 2021, 13, 2139). As an indispensable resource, albumin-based drug delivery system has demonstrated potential value in countering NSCLC.” This is critical to address in this manuscript, the authors should enrich this part in the revised version.

Authors’ answer: We thank the reviewer for showing us these details that need clarification. The suggested sentences were included in the introduction. Refers to lines 89-98

3: An additional check of the language would be beneficial.

Authors’ answer: We thank the reviewer for the suggestion. The manuscript was checked by a native English-speaking colleague.

4: More recent research information and references on “combination drug therapies” should be added.

Authors’ answer: We thank the reviewer for suggesting adding more information about this topic because it helped strengthen the article. The paragraph now reads as follows: The term combination therapy is used when two or more drugs positively affect the drugs’ pharmacodynamics interaction (i.e., additive and synergistic). The primary purpose of combination therapy is to lower drug doses so that the patients experience fewer side effects while the efficiency of the treatment is enhanced [59]. Furthermore, repurposing old drugs with new applications as a potential treatment for other diseases, such as lung cancer, is advantageous to the drug combination [60]. In recent studies, drug combination therapy incorporating a natural-derived product have shown beneficial effect against NSCLC [61]. The revealed results included cell viability reduction, apoptosis promotion, and protein expression downregulation to delay chemoresistance occurrence in lung cancer [62] and inhibition of cancer cell proliferation [63]. In our experiments, a synergistic effect is revealed in the drug combination of Dox and BeA, increasing the cytotoxic effect. This synergy was demonstrated by co-incubating A459 cells with free Dox and BeA, and the assistance of the nanosized BSA carrier increased this effect raised.

Refer to lines 585-597.

Reviewer 2 Report

The paper constitutes a well-prepared, well-organized and really interesting work which is definitely worth considering for publication. The only minor revisions need to be as follows:

1) Section 2.3.1.: the methodology applied to determine the BSA protein concentration in the DDS should be described in more detail.

2) Section 2.3.4.: why Authors selected the environments with pH = 6.8 and 7.4 to perform in vitro release studies? This should be explained.

3) Figure 1.: "wavelenght" in the description of x axis should be replaced by "wavelength".

4) Final conclusions need to be supplemented with some quantified data.

5) Section References should be consistent and prepared in line with the requirements of the Journal, e.g. the whole journal names should be replaced by their abbreviations. 

Author Response

The authors acknowledge and thank for all the comments and suggestions made by reviewers. The incorporation of these changes helped us strengthen and clarify the article. All the changes made by the authors were highlighted for the convenience of the reviewers. We answered what we have done in each reviewer’s comment, referring to the sections and parts previously commented on by reviewers.

Reviewer #2

(Reviewer’s comments in italics for better understanding)

The paper constitutes a well-prepared, well-organized and really interesting work which is definitely worth considering for publication. The only minor revisions need to be as follows:

1) Section 2.3.1.: the methodology applied to determine the BSA protein concentration in the DDS should be described in more detail.

Authors’ answer: We thank the reviewer for suggesting adding more information about this measurement because it helps strengthen the methodology. The paragraph now reads as follows: We used spectrophotometric measurements to determine each component’s concentration in the DDS. The standard protocol using the Bradford Coomassie reagent was utilized to determine the BSA protein concentration in the DDS from the absorbance at 570 nm [30]. A calibration curve was constructed with BSA standards. An aliquot of 5 µl was analyzed spectrophotometrically of each sample and BSA standard following the Bradford Coomassie protocol, which established that 150 µl of Bradford reagent must be added to each sample and standard. Then, the samples were incubated for 10 minutes at 25 ºC in the dark and followed by absorbance measurement at 570 nm.

Refer to lines 136-144.

2) Section 2.3.4.: why Authors selected the environments with pH = 6.8 and 7.4 to perform in vitro release studies? This should be explained.

Authors’ answer: We thank the reviewer for asking us these details that need clarification. The paragraph now reads as follows: Drug release experiments were conducted as described by Morales-Cruz et al. [33]. Briefly, one ml of a 5 mg/ml solution of the DDS in 25 mM sodium bicarbonate at pH 6.8 and 7.4 (to mimic tumor microenvironment and physiological conditions, respec-tively), was incubated at 37 ºC under stirring. At pre-determined time intervals (typi-cally 24 h), the DDS sample was filtered through a centrifugal filter device at 12,500 rpm for 15 min. The filtrated solution (with the released drug) was measured at 470 nm (Dox intrinsic absorbance) after the filtration through the centrifugal filter device. Also, the BeA component was quantified from the filtered solution using the vanillin-sulfuric acid assay as described in section 2.3.1. The DDS was re-suspended in 1 mL of bicarbonate buffer to maintain sink release conditions. The data are presented as accumulative release profiles of an average triplicate, with standard deviations.

Refer to lines 171-183.

3) Figure 1.: "wavelenght" in the description of x axis should be replaced by "wavelength".

Authors’ answer: We thank the reviewer for correcting this typo. Refer to lines 364.

4) Final conclusions need to be supplemented with some quantified data.

Authors’ answer: We thank the reviewer for suggesting adding more information because it helped strengthen the final conclusion. The paragraph now reads as follows: In this study, we developed novel BSA-based NPs  with the capacity to incorporate lipophilic (i.e., BeA) and hydrophilic moieties (i.e., Dox). The BSA(BeA+Dox) DDS has the characteristics of size and charge (181 ± 2 nm and -2.1 ± 0.7 mV, respectively) suitable for passive delivery into lung tumors and to be internalized by NSCLC cells. The surface charge of the DDS NP was slightly negative, and this property has been shown to prolong circulating half-life and accumulation in tumors. Besides the synergistic effect of the BeA and Dox combination, BSA(BeA+Dox) DDS diminished to half the expression of EGFR commonly produced in NSCLC by chemotherapeutic agents such as Dox. BSA(BeA+Dox) has the potential to increase the bioavailability of Dox and BeA and decrease chemoresistance during intravenous administration. Further studies of drug combination DDS must be continued to explore anticancer efficacy for in vivo studies.

5) Section References should be consistent and prepared in line with the requirements of the Journal, e.g. the whole journal names should be replaced by their abbreviations.

Authors’ answer: We thank the reviewer for correcting the Reference section. The reference section now is prepared in line with the requirements of the Journal.